# Recombinant expression and preliminary characterization of Peptidyl-prolyl cis/trans-isomerase Rrd1 from *Saccharomyces cerevisiae*

**Mohd Kashif**[1]*, **Ahad Amer Alsaiari**[2], **Bhupendra Kumar**[1], **Mohd Asalam**[3], **Mohammad Imran Khan**[4], **Abrar Ahmad**[4], **Rayees Ahmad Lone**[1], **Mazen Almehmadi**[2], **Mazin A. Zamzami**[4], **Mohd Sohail Akhtar**[3]

**1** Center for Plant Molecular Biology and Biotechnology Division, CSIR-National Botanical Research Institute, Lucknow, Uttar Pradesh, India, **2** Department of Clinical Laboratory Sciences, College of Applied Medical Sciences, Taif University, Taif, Saudi Arabia, **3** CSIR-Central Drug Research Institute Sector 10, Jankipuram Extension, Lucknow, Uttar Pradesh, India, **4** Department of Biochemistry, King Abdulaziz University, Jeddah, Saudi Arabia

* kashifibu@gmail.com

**Data Availability Statement:** All relevant data are within the paper and its Supporting information files.

## Abstract

*Sacchromyces cerevisiae* Peptidyl-prolyl cis/trans-isomerase Rrd1 has been linked to DNA repair, bud morphogenesis, advancement of the G1 phase, DNA replication stress, microtubule dynamics and is also necessary for the quick decrease in Sgs1p levels in response to rapamycin. In present study, Rrd1 gene was amplified by standard PCR and subsequently cloned downstream to bacteriophage T7 inducible promoter and lac operator of expression vector pET21d(+). Additionally, immobilized metal affinity chromatography (IMAC) was used to purify the protein upto its homogeneity, and its homogeneous purity was further confirmed through western blotting. Size exclusion chromatography implies that Rrd1 is existing as monomer in its natural state. Foldwise Rrd1 protein belongs to PTPA-like protein superfamily. Rrd1 showed characteristic negative minima at 222 and 208 nm represent protein typically acquired α helix in the far-UV CD spectra. Fluorescence spectra showed properly folded tertiary structures of Rrd1 at physiological conditions. Rrd1protein can be identified from different species using a fingerprint created by PIPSA analysis. The protein's abundance could aid in its crystallization, biophysical characterization and identification of other-interacting partners of Rrd1 protein.

## Introduction

Peptidyl-prolylcis/trans-isomerase Rrd1 work as major subunit of the Sit4p-Tap42p—Rrd1p complex and as activator of the PP2Aphosphotyrosyl phosphatase activity. It is involved in the advancement of the G1 phase, DNA repair, dynamics of microtubule and bud morphogenesis [1, 2]. It is also necessary for the rapid reduction of Sgs1p levels in response to two genes, YPA1 (Rrd1- Rapamycin- resistant deletion) and YPA2 (Rrd2), in budding yeast encodes the protein PTPA [3]. Individual YPA1 deletion results in a more rigorous phenotypic expression than individual YPA2 loss. Deletion of Rrd1 (YPA1) causes growth abnormalities, rapamycin

**Funding:** The Deanship of Scientific Research (DSR) at King Abdul Aziz University, Jeddah, Saudi Arabia has funded this project, under grant no. (KEP-25-130-42), which is greatly acknowledged.

**Competing interests:** The authors declare that they have no competing interests.

resistance, irregular actin distribution and abnormal bud shape [3]. Nutrient sensing and signaling network that regulates number of growth-activities in response to growth factor signaling and nutrient availability is TOR. Deletions YPA1 and YPA2 confer resistance of rapamycin most likely by indirect regulation of TOR [4]. Rrd1 shares 35% sequence similarity with the hPTPA (human phosphotyrosyl phosphatase activator) and is well conserved among the eukaryotes [5, 6]. Rrd1 previously shown to be enhance the type 2AThr/ Ser phosphatase PP2A poor phosphatase activity. Rrd1 can physically known to associate with a type of Thr/ Ser phosphatase Sit4, a well characterized phosphatase that is similar to PP2A [7]. Rapamycin drug well known to interacts with peptidyl-prolyl cis/trans isomerase Fpr1 in *S. cerevisiae*. This protein-drug complex led to inactivation of the Tor1 kinase, leading to G1 growth arrest and a significant alteration de the transcriptional profile [8, 9]. By virtue of its separation from the well characterized inhibitor complex Tap42-Sit4, Sit4 is activated when Tor1 is inhibited. Rapamycin well known anticancer and immunosuppressant drug that works by blocking the TOR signaling system [10, 11]. Rrd1 gene is necessary for the strong transcriptional response that rapamycin mediates in yeast. Rrd1 and RNA polymerase II (RNAPII) co localize on genes that are actively transcribed and recruited to genes that are rapamycin responsive [12, 13]. Surprisingly, in absence of Rrd1, RNAPII continues to connect improperly with a number of ribosomal genes and is unable to recruit responsive genes to rapamycin [14]. Rrd1 modifies the phosphorylation level of RNAPII C terminal domain in this process, which happens in parallel of the recruitment of TATA box interacting proteins [15]. In addition to rapamycin, Rrd1 is also implicated in a number of other transcriptional related stress activities. Rrd1 is a new type of transcription elongation factor that regulates RNAPII response to stress during transcription. Rrd1/PTPA might activate PP2As byway of this PPIase activity. Although this is not the case for other PPIases, the in vivo target and biological purpose of Rrd1's PPIase activity have not yet been determined. Recent reports showed that Rrd1 interacts with RNAPII and isomerizes Rpb1's CTD both *in vivo* and *in vitro* condition [16]. *Saccharomyces cerevisiae*' Peptidyl-prolylcis/trans-isomerase Rrd1 has not yet been biophysically described. *Escherichia coli* protein expression enables the quick production of significant quantities of recombinant proteins [17]. One of the most popular is pET-21d (+) vector for recombinant protein expression and purification [18]. In the current study, we showed the cloning of the *rrd1* gene in pET-21d (+) effective overexpression and purification in *E. coli*. Production of the recombinant protein can be used to obtain its biophysical characterization and to identify novel-associating partners of Rrd1 protein.

## Materials and methods

Restriction enzymes and Deoxynucleotide triphosphates (dNTPs) were procured from New England Biolabs (NEB), United States. All other chemicals and reagents were of the greatest purity and were bought from (Sigma-Aldrich), Mumbai, India and Sisco Research Laboratories. The nickel-nitrilotriacetic acid (Ni-NTA) agarose matrix was acquired from Qiagen, CA, USA. The Himedia Laboratories in Mumbai, India provided the bacterial culture media.

### *S. cerevisiae* genomic DNA isolation

The genomic DNA of the *S. cerevisiae* (BY4741 strain) (MATa his3Δ1 leu2Δ0 met15Δ0 ura3Δ0) was manually extracted using a method described elsewhere [19].

### Cloning

*S. cerevisiae* Rrd1 gene was amplified by using PCR. The gene-specific primers 5'-GATGCTAGCATGTCTCTGGATCGT-3' (forward primer) and

5'GCCCTCGAGTCTACGTAGTCTATCTCTTG –3' (reverse primer) with NheI and XhoI restriction sites respectively. The following cycle parameters were used in the PCR amplification with Taq DNA Polymerase (New England Biolabs, USA). Initial Denaturation_95 (2 min) Denaturation_95 (30 Sec) Annealing_58 (30 Sec) Extension_72 (1min (2kb/min)) Final Extention_72 (5 min) No of Cycles_40, make up the initial denaturation temperature an increase in the initial denaturation temperature [20, 21].

## Overexpression and purification of the recombinant Rrd1 protein

Ampicillin (100 μg/mL) added on LB agar plates, recombinant cells harbouringpET21d (+)-Rrd1 plasmid construct were examined. Positive clones were selected and grown in 5 mL of LB broth ampicillin (100 μg/mL). Two varying IPTG concentrations (0.5 mM and 1.0 mM) with varying temperatures were examined to maximize the protein expression of Rrd1. Four separate expression systems C41 BL-21, BL21-CodonPlus and Tuner were employed to express the protein. For large scale protein production, sterile LB broth (400 mL) containing ampicillin (100 μg/mL) and 1 percent primary culture was inoculated. Furthermore cultures were then incubated in shaker incubator at 37˚C (180 rpm) shaking until the $OD_{600}$ reached upto 0.7, after which culture was induced with IPTG (0.5 mM) concentrations for overnight (12 hrs). Centrifugation at 4˚C was done to pellet down overnight induced and grown cells. Additionally pellet was resuspended in 20 mL resuspension buffer (300 mM NaCl, 50 mM Tris) and PIC (protease inhibitor cocktail) (pH 8.0). Sonication was performed with 26 percent amplitude for 10 s pulse on and 15 s pulse off for 20 minutes, culture cells were lysed on ice. Centrifugation of crude lysate took place at 4˚C for 20 min at 13,000 rpm. Prior to affinity chromatography the supernatant was filtered through (0.45 m pore PVDF membrane filter). The matrix was washed with 10 bed volumes of equilibration buffer (50 mM Tris (pH 8.0), 300 mM NaCl). The Ni-NTA Agarose matrix (Qiagen, USA) was loaded with the protein supernatant. Additionally; Rrd1 was homogeneously purified using Ni-NTA in accordance with manual guidelines. Bradford technique was used to determine the concentration of recombinant protein [20, 22].

## AKTA FPLC / size exclusion chromatography (SEC)

AKTA FPLC was used to implement on a SuperdexTM200 10/300 GL column with (20 mM Tris, pH 8.0, and 150 mM NaCl) at room temperature was used to measure the molecular weight of recombinant Rrd1. Standard molecular weight markers were used to pre-calibrate the column. Dialyzing the Rrd1 protein in the preferred buffer pH 7.0. With flow (0.4 ml/min) and with detection at 280 nm, the columns were washed and pre-equilibrated suitable buffer at 25˚C [23].

## Western blot analysis

12 percent polyacrylamide gel was used to transfer the purified recombinant Rrd1 protein PVDF membrane for 90 minutes at 50 mV. With 5% skim milk, the protein-transfer membrane was stopped for three hours. After that, the membrane was cleaned and treated with a Penta-His Antibidy, BSA free (QIAGEN) 1:1000 (100 ng/ul) in PBS contain 1% BSA. The membrane washed then treated with Secondary Antibody (Promega Corporation) Goat Anti-Mouse IgG HRP conjugate (1mg/ml) washed three times with PBS, pH 7.4 PBST (PBS,.05 percent Tween). Chromomeric Substrate (HRP Conjugate) (Invitrogen Novex HRP Chromogenic Substrate) was used to develop the membrane [22, 24].

## Spectro-fluorophotometery

Optimal protein concentration (2–4 μmole) was taken and CD spectra were captured in a quartz cell using a spectro-fluorophotometer from the CSIR CDRI, Lucknow. At normal temperature, 1.0 cm path length fluorescence spectra were captured. Before recording the fluorescence data, the protein sample was equilibrated for 30 minutes. The emission was measured (between 300 and 400 nm). The excitation wavelength being set at 280 nm [25, 26].

## Circular dichroism (CD)

Spectropolarimeter was used to measure the circular dichroism (CD) spectra (CSIR CDRI, Lucknow). Ammonium D-10 Camphor sulfonate was used to precalibrate the instrument. For scanning between 250 and 195 nm, (path length of 0.1 cm) was implemented. While for scanning between 300 and 250 nm, (path length of 1 cm) was used. The mean residue ellipticity (MRE) in $deg.cm^2.dmol^{-1}$

$$\mathbf{MRE} = \frac{\mathbf{\Theta}_{obs}(\mathbf{mdeg})}{10 \times n \times l \times C_P}$$

Where n is the number of peptide bonds per molecule, l, the length of the light path (in centimetres), Cp is molar concentration and øobs, is the observed ellipticity (CD) in milli-degrees [27, 28].

## Sequence alignments and phylogenetic analysis

The patterns found in sequence logos and multiple sequence alignment is now frequently represented graphically using PDBsum server and Esprit3.0 [29, 30]. Neighbor-Joining (NJ) trees were built for phylogenetic analysis using data from the National Center for Biotechnology Information (NCBI), U.S. National Library of Medicine. It facilitates the completion of phylogenetic histories, sequence alignments, and molecular phylogenetics [31].

## Homology modeling and PIPSA analysis

Rrd1 protein has an entry in the S.G.D. (*Saccharomyces cerevisiae* genome database) with the SGD ID S000001492. Rrd1 of *Saccharomyces cerevisiae* had known structure. We made an effort to build *Saccharomyces cerevisiae* Rrd1 entire 3D structure [32]. First, the RCSB-PDB (protein data bank) and Phyre2 [33] were searched for suitable templates using the BlastP programme. Through the assistance of the web programme Phyre2, the yeast Rrd1 protein's structure was created using the PDB structure 2IXP as a template. The model visualization and columbic surface analysis were done using UCSF Chimera [34]. PDB files with the protein coordinate were necessary for the PIPSA (Protein Interaction Property Similarity Analysis) online analysis. These can fulfill user-supplied and PDB identification code-specified objectives. Both empirically derived structures from the relevant databases like MODBASE, or comparative modelling techniques utilising MODELLER or SWISSMODEL, can be used to build user-supplied structures. Basically, the webPIPSA uses the electrostatic potentials of a number of proteins ($kcal\ mol^{-1}\ e^{-1}$) to calculate similarity indices [35, 36].

# Results

## PCR amplification and cloning of the *rrd1* gene

Gene-specific primer set with NheI and XhoI restriction site was used and 1.1 Kb Rrd1 gene amplified by conventional PCR before being cloned into the vector pET-21d (+) (Novagen).

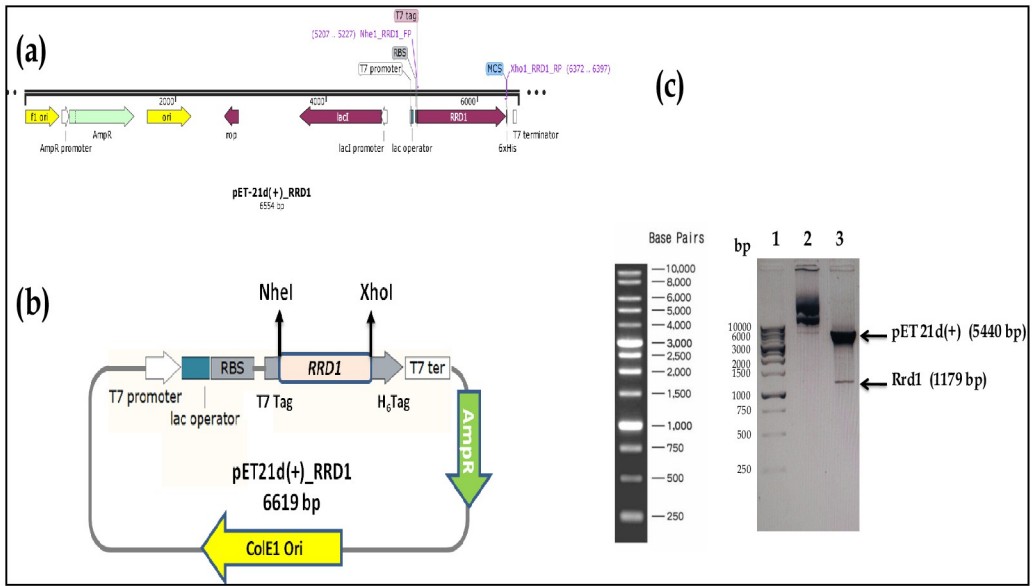

**Fig 1. Cloning of Rrd1.** Confirmation of clones by double digestion as analyzed by agarose gel electrophoresis. (a) Schematic vector linear map of pET21d (+)-Rrd1 construct. (b) Schematic vector circular mapofpET21d (+)-Rrd1 construct. (c) Lane-1 represents marker of 1kb DNA Ladder (BM010-R500,BR Biochem) and lane-2 undigested pET21d (+)-Rrd1 plasmid construct, lane-3 represents double digested clone having vector and insert.

Clone was verified through double digestion using the restriction enzymes NheI and XhoI (Fig 1c lane 3). Two bands corresponding to the vector backbone (5.4 kb) and the rrd1 gene (1.1 kb), were seen after double digestion respectively (Fig 1b and 1c lane 3). Sequencing of positive clones was done, and the Rrd1 sequence from the SGD (*Saccharomyces* Genome Database; http://www.yeastgenome.org) was compared. Rrd1 sequence that is accessible in SGD and the clone sequence were 99.99% identical. The most favoured way for mass synthesis of recombinant proteins is still *Escherichia coli*. The solubility of heterologous proteins and their purification with affinity chromatography have been significantly improved by the introduction of fusion proteins.

## Optimization of recombinant Rrd1 expression and solubility

Four distinct expression systems of *E.coli* C41, BL-21, BL21-CodonPlus and Tuner strain were employed to express recombinant Rrd1. Although Rrd1 express in ample amount in all the strain but its expression was observed maximum in C41 strain (Fig 2a lane 3). Two different (0.5 mM and 1.0 mM) IPTG concentrations examined exhibited nearly identical levels of Rrd1 protein expression. Therefore, a concentration of 0.5 mM IPTG is thought to be ideal for Rrd1 expression. Rrd1 protein solubility optimization done after sonication. The outcomes demonstrated that the recombinant protein product has a molecular weight of approximately 46.5 kDa (45 kDa Rrd1+ 0.85 kDa His tag+1.1 kDa T7 tag), which is consistent with the projected size of pET21d (+)-Rrd1. Rrd1 protein gets solublized properly that indicates feasibility of its purification. In the soluble fraction, Rrd1 protein predominated (Fig 2b, lane 5).

Rrd1 protein was purified through affinity purification through Ni-NTA column followed by gel filtration chromatography. Total soluble protein (TSP) after sonication contains ~15mg/mL proteins, while Rrd1 yield after purification was ~0.3 mg/mL. The single protein band on SDS-PAGE which corresponds to a molecular mass of 45 kDa, showed that the

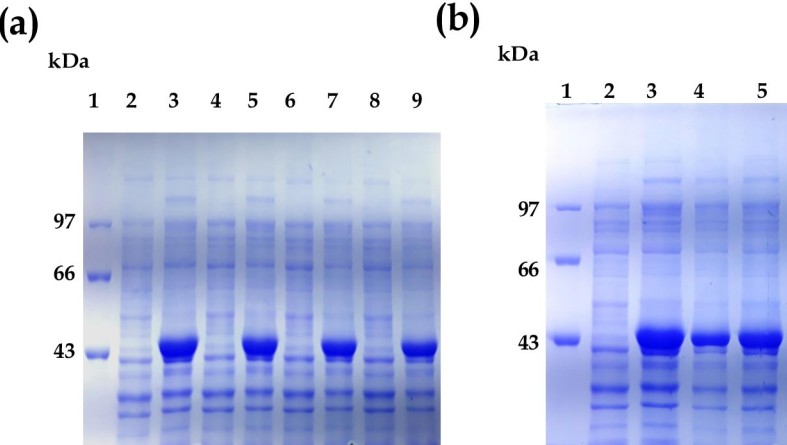

**Fig 2. Overexpression in various expression strain and solubility optimization of Rrd1.** (a) Overexpression of Rrd1. Lanes 1–9 represents protein molecular weight marker, uninduced sample and sample after overnight induction in C41, BL21, BL21-CodonPlus and Tuner strain respectively. (b) **Solubility optimization of Rrd1**. Lanes 1–5 represents protein molecular weight marker, uninduced sample, sample after overnight induction, pellet (insoluble fraction) and soluble fraction after sonication in C41 strain.

purified protein was homogeneous (Fig 3a, lane 6). The single protein band on western blot analysis which correspond to a molecular mass of 45 kDa, showed that the homogeneously purified Rrd1 protein (Fig 3b, lane 6). AKTA-FPLC on a SuperdexTM200 10/300 GL column with (20 mM Tris, pH 8.0, and 150 mM NaCl) at room temperature was used to measure the oligomeric conformation and molecular weight of recombinant Rrd1. When compared to the common molecular weight marker, the protein eluted at a volume of 15.4 ml, which is equivalent to around 45 kDa (Fig 3c and 3d). This implies that Rrd1 is existing as monomer in its natural state (Fig 3c).

## Primary sequence analysis and homology modeling of Rrd1

PDBsum server showed sequence coloured by residue conservation and secondary structure of the protein. Protein surface showing most poorly (blue) while highly conserved (red) regions (Fig 4a). This concludes that the majority of the Rrd1 protein amino acids are extremely well conserved throughout evolutionary history (Fig 4a). The Rrd1 protein model that is purely based on a sequence alignment made by HMM-HMM matching was predicted using SWISS MODEL and Phyre2. The secondary structure of the recombinant Rrd1 protein was predicted using the template known secondary structure (2XIP). Basically, topological secondary structure of Rrd1 protein is made up of eight to nine alpha helices according to Phyre2 secondary structure analysis (Fig 4a, S2 Fig in S1 File). Through the usage of UCSF chimera, Rrd1 columbic surface (Phyre2, Model dimensions Model dimensions (Å):X:59.504 Y:56.518 Z:51.967) was coloured using template (2XIP). Fold wise Rrd1 protein belongs to PTPA-like protein superfamily [37].

## Conserved domain of Rrd1protein

CDD server (Conserved domain) online tool was utilized to determine the Rrd1 recombinant protein's domain [38]. The Rrd1 proteins from *Saccharomyces*, *Arabidopsis thaliana* and *H. sapiens*, *Drosophila melanogaster* were shown through multiple sequence alignment to have well-defined and conserved Rrd1 protein belongs to PTPA-like protein superfamily (Fig 5, S1

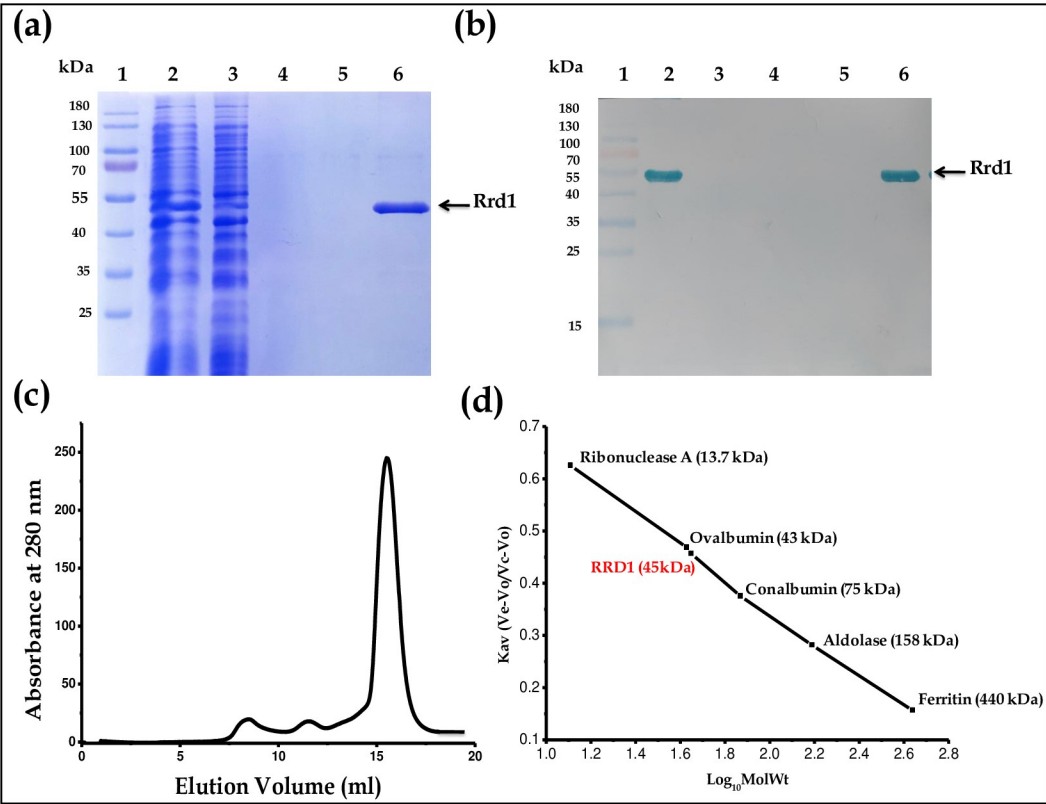

**Fig 3. Affinity purification and size exclusion chromatography of Rrd1 protein.** (a) Lanes 1–6 represents protein marker, lysate, sample after sonication, flow through, wash buffer1, wash buffer2 and eluted fraction from Ni NTA column respectively. (b) Western blot analysis of Rrd1. Lanes 1–6 represents protein marker, lysate, sample after sonication, flow through, wash buffer1, wash buffer2 and eluted fraction from Ni NTA column respectively. (c) The size exclusion chromatography profiles of purified Rrd1 (with flow rate of 0.4 ml/min and detection at 280 nm) (d) The graph where the column was calibrated with standard molecular mass markers.

Fig in S1 File). PPIases are well known to catalyze the folding of proteins and also the cis-trans isomerization of proline imidic peptide bonds in small proteins [39, 40]. It also acts as a controlling subunit for threonine/serine PP2A (protein phosphatase 2A) modulating its substrate specificity or activity, led to a conformational alteration in the catalytic subunit. They all possess Protein phosphatase 2A (PP2A) phosphatase activator is another name for phosphotyrosyl phosphatase activator (PTPA) [41].

## Circular dichroism and fluorescence spectroscopy of Rrd1 protein

Fluorescence spectroscopy is frequently used to examine structural changes brought on by changes in temperature, pH, ionic strength, solvent, and ligands in conjugated systems, aromatic molecules, and stiff planar structures. The tryptophan fluorescence is frequently employed as a tool to track alteration in proteins and draw conclusions about their local dynamics and structure [42]. The preferred method for analyzing chiral compounds in solution, particularly biologically significant chemicals including proteins, nucleic acids, carbohydrates and pharmaceuticals is circular dichroism (CD) spectroscopy [43]. There are eight tryptophan residues present in Rrd1. Consequently, CD analysis and protein fluorescence can be utilized to determine a protein structural state. The CD spectra and fluorescence

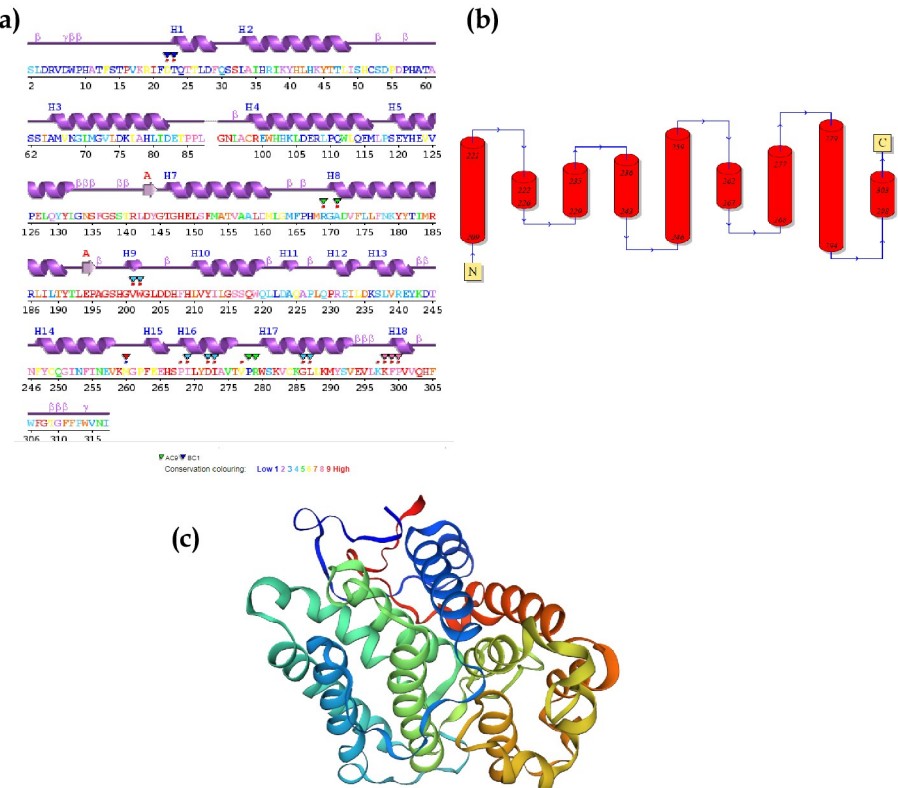

**Fig 4. Sequence conservation, alignments and homology modeling of recombinant Rrd1.** (a) PDBsum gives an idea of sequence conversation and multiple sequence alignment. Protein surface showing most poorly (blue) and highly conserved (red) regions. (b) Topological diagram showing secondary structure of Rrd1 protein. (c) Template (2IXP) was identified from Phyre2 and SWISS-MODEL analysis and implemented to predict secondary structure of Rrd1 protein, generated through homology modeling.

spectra both revealed appropriately folded secondary and tertiary structures of recombinant Rrd1 at physiological pH respectively (Fig 6a and 6b). Existence of prominent negative minima (at bands at 222 nm and 208 nm), which represents that Rrd1 have typical α helical protein in the far-UV CD spectra (Fig 6a). Fluorescence emission spectra show a peak (max 337) that corresponds to the correctly folded tertiary structure of recombinant Rrd1 protein (Fig 6b).

## Protein Interaction Property Similarity Analysis (PIPSA)

The comparison of the various electrostatic interaction characteristics of proteins is made possible via the PIPSA online service. It enables the classification of proteins based on the characteristics of their interactions. PIPSA can help with enzyme kinetic parameters, estimation of binding characteristics and function assignment. PIPSA analysis performs on various protein sites and to compare the binding characteristics of user-defined protein groups. The clustering of proteins in one group at the graph, which denotes their evolutionary close relation, is due to their nearly identical electrostatic properties. The EP (electrostatic potentials) of Rrd1-(like proteins) from four distinct species was compared using the webPIPSA service. Protein from species with extremely similar electrostatic potentials including *Saccharomyces cerevisiae* (2IXN), *S. aureus* (3ROF) and *Homo sapiens* (2IXM), are found in one distinct subcluster,

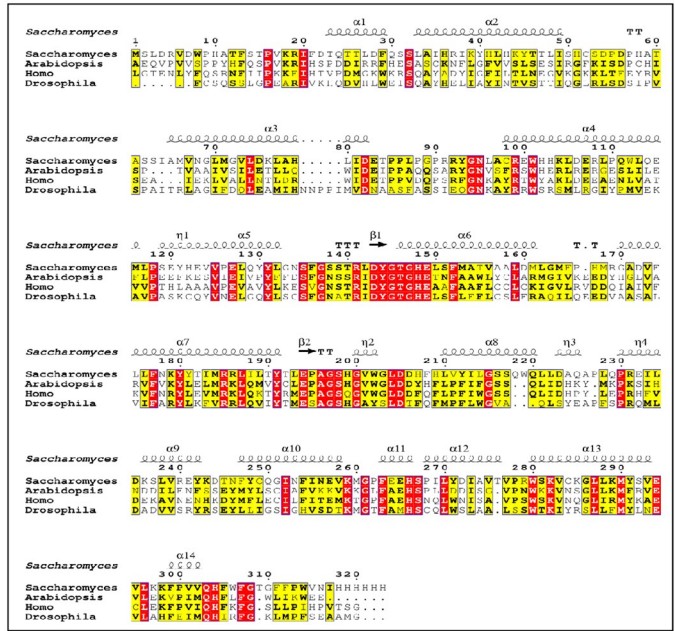

**Fig 5. Multiple sequence alignment of *Saccharomyces*, *Arabidopsis thaliana H. sapiens* and *Drosophila melanogaster* Rrd1 proteins.** (A) Alignment was performed through ClustalW followed by Espript3.0 analysis. Shown through multiple sequence alignment to have well-defined and conserved Rrd1 protein belongs to PTPA-like protein superfamily.

while *Escherichia coli* (1TE2) is found in a separate second subcluster (Fig 7, S3 Fig in S1 File). This will give us more knowledge on the structural and physico-chemical characteristics of Rrd1 like protein specific regulation. Insights into isoform-specific regulation are also provided by webPIPSA, which are not possible through study at the sequence level alone.

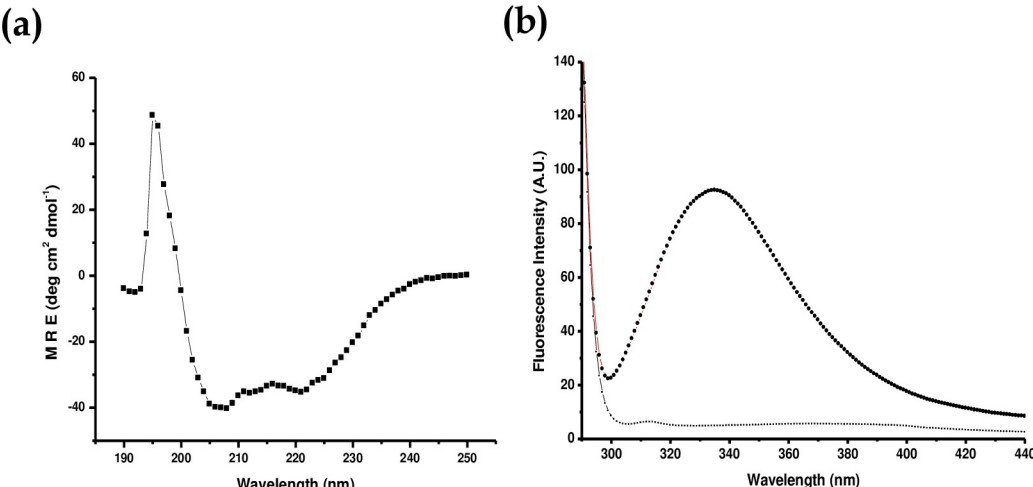

**Fig 6. Secondary and tertiary structure analysis of recombinant Rrd1.** (a) CD spectra of Rrd1. Rrd1 have typical six α helix with characteristic negative minima at bands at 222 and 208 nm. (b) Fluorescence emission spectra (λmax- 337) of Rrd1 protein.

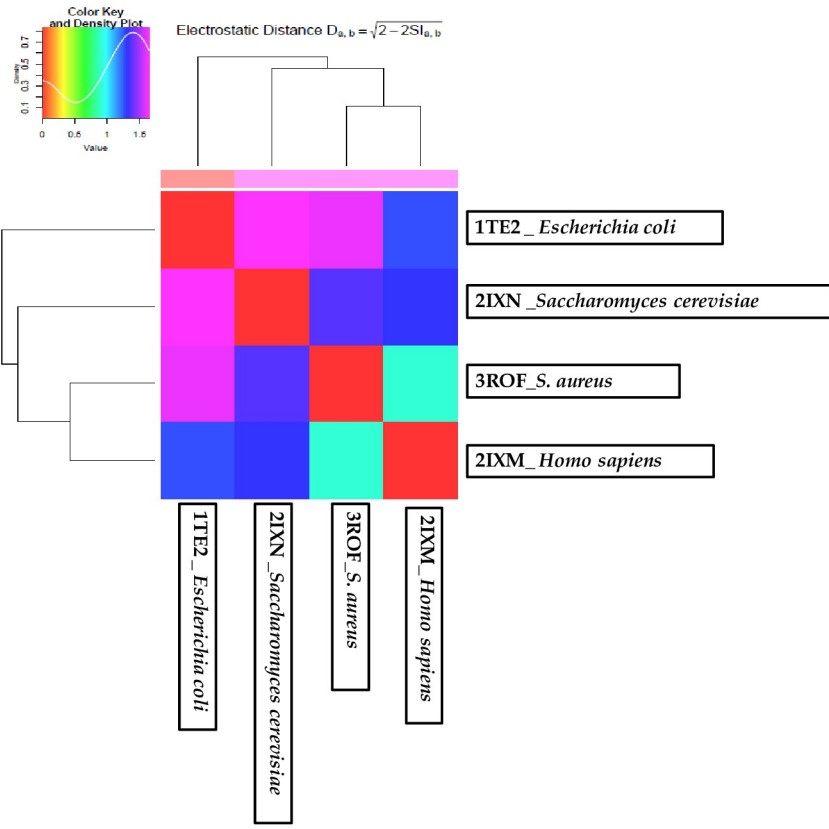

**Fig 7. The EP (electrostatic potentials) of Rrd1-(like proteins) from four distinct species was compared using the webPIPSA service.** Protein from four species with extremely similar EPs, including *Saccharomyces cerevisiae* (2IXN), *S. aureus* (3ROF) and *Homo sapiens* (2IXM) are found in one distinct subcluster, while *Escherichia coli* (1TE2) is found in a separate second subcluster.

## Phylogenetic analysis

Another viewpoint on biodiversity is offered by phylogeny, which enables a methodical comparison of the diversity and uniqueness of taxa. Phylogenetic trees are typically employed to assess species richness in concordant groups, despite the fact that numerous particular metrics of phylogenetic diversity have been presented. The two groups shared a common ancestor more recently the closer they are to one another geographically [44]. Rrd1 protein belongs to PTPA-like protein superfamily (Fig 8). PPIases are well known to catalyze the folding of proteins and also the cis-trans isomerization of proline imidic peptide bonds in small proteins [45]. It also acts as a controlling subunit for threonine/serine PP2A (protein phosphatase 2A) modulating its substrate specificity or activity, led to a conformational alteration in the catalytic subunit. They all possess Protein phosphatase 2A (PP2A) phosphatase activator is another name for phosphotyrosyl phosphatase activator (PTPA) (Fig 8).

## Discussion

Rrd1 protein belongs to PTPA-like protein superfamily that catalyzes the folding of proteins and also the cis-trans isomerization of proline imidic peptide bonds in small proteins. Peptidyl-prolyl cis/trans-isomerase Rrd1 is also work as a major subunit of the Tap42p-Sit4p-Rrd1p

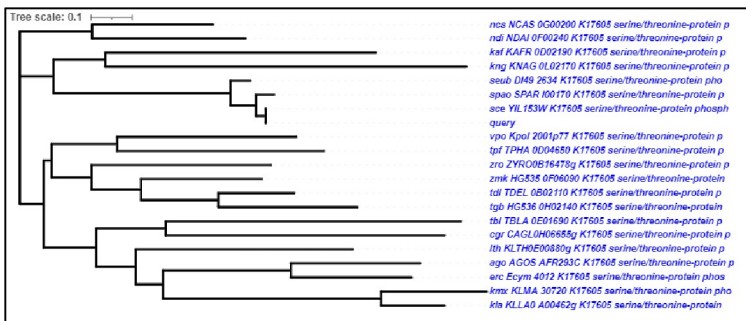

**Fig 8. Phylogenetic analysis of Rrd1 protein superfamily.** Neighbor-Joining Phylogenetic tree of protein PTPA/Rrd-like protein superfamily. Using the full-length sequence related organisms (20) having PTPA/Rrd-like protein, tree was constructed with the NCBI based phylogenetic analysis.

complex and work as activator of the PP2A phosphotyrosyl phosphatase activity. It is involved in the advancement of the G1 phase, DNA repair, dynamics of microtubule and bud morphogenesis. Well known tumour p53 (a kind of suppressor protein) can hinder PTPA expression level by an unknown mechanism that negatively regulates YY1 [45]. Gene *rrd1* was amplified and cloned downstream to bacteriophage T7 inducible promoter and lac operator of expression vector pET21d (+). Additionally, protein was expressed and purified upto its homogeneity and this was further confirmed through western blotting using anti-His antibody. Size exclusion chromatography implies that Rrd1 is existing as monomer in its natural state. Rrd1 is a member of the PTPA-like protein superfamily. PDBsum server showed sequence coloured by residue conservation and secondary structure of the protein. Protein surface showing most poorly conserved (blue) while highly conserved (red) regions this conclude that the majority of the Rrd1 protein's amino acids are extremely well conserved throughout evolutionary history. The Rrd1 protein model was predicted using SWISS MODEL and Phyre2. Structure of the recombinant Rrd1 protein generated using the template known structure (2XIP). Basically, fundamental structure of Rrd1 protein is made up of eight to nine alpha helices according to Phyre2 secondary structure analysis. Phyre2 model dimensions has been reported (Å): X:59.504 Y:56.518 Z:51.967). Rrd1 from yeast has a conserved PTPA domain. Rrd1 have eight tryptophan residues and under physiological conditions, protein showed appropriately folded protein tertiary structure in fluorescence spectra. The prominent negative minima (at the 222 and 208 nm bands) in the far-UV CD spectra of Rrd1 basically contains α helix. These α -helix proteins have been well documented in biological systems including transmembrane proteins (G protein-coupled receptors (GPCRs)) [46] and zinc finger motifs (a kind of DNA binding proteins) [47]. PIPSA was implemented to compare the electrostatic characteristics of Rrd1-like proteins. On the basis of PDB structure, PIPSA analysis gives idea regarding the EPs. Protein from species with extremely similar electrostatic potentials, including *Saccharomyces cerevisiae* (2IXN), *S. aureus* (3ROF) and *Homo sapiens* (2IXM), are found in one distinct subcluster, while *Escherichia coli* (1TE2) is found in a separate second subcluster. The interaction characteristics of several protein families, including human RabGTPase proteins [48] and blue copper proteins [49] have previously been investigated using this method.

## Conclusions

Rrd1 involved in a number of vital process like advancement of the G1 phase, DNA repair, dynamics of microtubule and bud morphogenesis. Since role of Rrd1 in cis-trans

isomerization, dynamics of microtubule, p53 (tumor suppressor protein) inhibition by PTPA expression, bud morphogenesis and DNA repair were still elusive in *S. cerevisiae* and humans, Furthermore, protein abundance could aid in its crystallization, biophysical characterization and identification of other-interacting partners of Rrd1 protein. Since Rrd1showed fair sequence conservation from yeast to humans, it is likely that this protein serves the same structural and functional role in humans.

## Supporting information

**S1 File.**
(DOCX)

**S1 Raw images.**
(PDF)

## Acknowledgments

I greatly acknowledged my supervisor, Dr. Mohd. Sohail Akhtar for his guidance and support. Ahad Amer Alsaiari, Assistant professor, College of Applied Medical science, Department of Clinical Laboratories Science, Taif university, Saudi Arabia greatly acknowledged for professional grammar and revising the language editing in this manuscript. BK acknowledged Academy of Scientific and Innovative Research (AcSIR), Ghaziabad, 201002, India.

## Author Contributions

**Conceptualization:** Mohd Kashif.

**Data curation:** Bhupendra Kumar, Mazin A. Zamzami.

**Formal analysis:** Mohd Asalam, Mazen Almehmadi.

**Funding acquisition:** Mazin A. Zamzami.

**Investigation:** Mohd Kashif, Mohammad Imran Khan.

**Methodology:** Mohd Kashif.

**Project administration:** Rayees Ahmad Lone.

**Resources:** Abrar Ahmad, Mohd Sohail Akhtar.

**Supervision:** Mohd Kashif.

**Writing – review & editing:** Ahad Amer Alsaiari, Mazin A. Zamzami.

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
