## [Decision Letter · Decision Letter 0]

28 Oct 2022

PONE-D-22-27039Recombinant expression and preliminary characterization of Peptidyl-prоlyl cis/trans-isоmerase Rrd1 from Saccharomyces cerevisiae.PLOS ONE

Dear Dr. Mohd Kashif,

Thank you for submitting your manuscript to PLOS ONE. After careful consideration, we feel that it has merit but does not fully meet PLOS ONE’s publication criteria as it currently stands. Therefore, we invite you to submit a revised version of the manuscript that addresses the points raised during the review process.Please submit your revised manuscript by December 12, 2022. If you will need more time than this to complete your revisions, please reply to this message or contact the journal office at plosone@plos.org. Please include the following items when submitting your revised manuscript:A rebuttal letter that responds to each point raised by the academic editor and reviewer(s). You should upload this letter as a separate file labeled 'Response to Reviewers'.A marked-up copy of your manuscript that highlights changes made to the original version. You should upload this as a separate file labeled 'Revised Manuscript with Track Changes'.An unmarked version of your revised paper without tracked changes. You should upload this as a separate file labeled 'Manuscript'.

We look forward to receiving your revised manuscript.

Kind regards,

Sabato D'Auria

Academic Editor

PLOS ONE

Reviewers' comments:

Reviewer's Responses to Questions

**Comments to the Author**

1. Is the manuscript technically sound, and do the data support the conclusions?

Reviewer #1: Partly

Reviewer #2: Partly

2. Has the statistical analysis been performed appropriately and rigorously? 

Reviewer #1: I Don't Know

Reviewer #2: N/A

3. Have the authors made all data underlying the findings in their manuscript fully available?

Reviewer #1: Yes

Reviewer #2: No

4. Is the manuscript presented in an intelligible fashion and written in standard English?

Reviewer #1: No

Reviewer #2: No

5. Review Comments to the Author

Reviewer #1: The manuscript by Mohd Kashif et al. entitled “Recombinant expression and preliminary characterization of Peptidyl-prоlyl cis/trans-isоmerase Rrd1 from Saccharomyces cerevisiae” describes the cloning of the rrd1 gene in pET-21d (+) its overexpression and purification in E. coli.

The authors conclude that the production of the recombinant protein can be used to obtain its biophysical characterization and to identify novel-associatin partners of Rrd1 protein.

I believe, as the authors say, that the production of the recombinant protein can be used to obtain its biophysical characterization and to identify novel-associatin partners of Rrd1 protein, but in my opinion the work should be improved.

I admit this manuscript with major revision.

In particular:

- I would like to suggest that the authors should indicate the yield of purification.

- I would suggest that the authors try to eliminate many repetitive parts in Discussion and Conclusion paragraphs.

- I suggest a revision of the English language.

Other points:

- Is the ampicillin concentration (50mg/mL) correct? May be 50 micrograms/milliliter.

- The resolution of the graphics is very bad. (Figure 1; Figure 3, C and D; Figure S1)

- In the “Overexpression and purification of the recombinant Rrd1 protein” paragraph, the parenthesis is missing (line 3).

- In the caption of Figure 2, opri,ization should be corrected

- The caption of Figure 3, should be corrected.

- The column flow is once declared to be 0.3 and once more 0.4 mL/min.

- The caption of Figure 4 and Figure 5 are not in italic.

- The font of the Figure 5 caption’s is different.

- The numerical sequence of the figures is wrong.

- To check punctuation.

Reviewer #2: The manuscript by Mohd Kashif et al. entitled “Recombinant expression and preliminary characterization of Peptidyl-prоlyl cis/transisоmerase Rrd1 from Saccharomyces cerevisiae” report cloning, expression and purification of the Rrd1 from Saccharomyces cerevisiae, and also an initial characterization. The authors declare that the protein is correctly cloned, expressed in the E. Coli strain, and purified. The authors, by the CD and the Spectro fluorescence, declare also that the protein is correctly folded.

The aim of this work is interesting (and this reviewer appreciates it), the paper is not well done in almost all parts, and the manuscript in the present form demands a deep revision before it can be published, so this reviewer suggests a major revision of the paper.

The paper should be modified in some parts.

Major issues:

1) The abstract needs to be revised according to the following suggestions.

2) The introduction section is too long and dispersive, nothing is reported on the aim of the paper. The results and the methods used should be removed from this section.

3) The section Materials and Methods describe also the results obtained. In this way is difficult to read and understand the paper, please split this section into two sections (e.g. M&M and Results).

4) About the purification is not reported the yield per gram of humid pellet, is not reported a spectroscopic analysis for the purity evaluation of the protein and for the quantization of the protein.

5) In the paragraph “Optimization of recombinant Rrd1 expression and solubility” the most important info is not reported “data not show”, why?

6) Some parts of the computational analysis should be reported before the expression and purification and the rest before the characterization.

7) In the paragraph “Primary sequence analysis and homology modeling of Rrd1” is reported (fig 3a) but it does not match with the text.

8) The paragraph “Structure analysis of Rrd1 protein” should be completely revised or removed. The CD spectra analysis is too simplistic, no spectra post-analysis was done. An investigation in the near-UV should be performed. By the fluorescence analysis performed (just steady-state spectra) is not possible to define the tertiary structure or the right folding of the protein, a deeper fluorescence analysis should be performed to declare the correct folding.

9) As a consequence of the point before also the Discussion and the Conclusions sections need to be revised.

Minor issue:

1) Please revise the text for some misspellings

2) Figure 3 panels c and d are in low resolution, please improve.

3) Figure S1 is in low resolution, please improve it.

6. PLOS authors have the option to publish the peer review history of their article (what does this mean?). If published, this will include your full peer review and any attached files.

Reviewer #1: No

Reviewer #2: No

---

## [Author Response · Author response to Decision Letter 0]

3 Feb 2023

Dear Dr. Mohd Kashif,

Thank you for submitting your manuscript to PLOS ONE. After careful consideration, we feel that it has merit but does not fully meet PLOS ONE’s publication criteria as it currently stands. Therefore, we invite you to submit a revised version of the manuscript that addresses the points raised during the review process.

We look forward to receiving your revised manuscript.

Kind regards,

Sabato D'Auria

Academic Editor

PLOS ONE

Reviewers' comments:

Reviewer's Responses to Questions

Comments to the Author

1. Is the manuscript technically sound, and do the data support the conclusions?

Reviewer #1: Partly

Reviewer #2: Partly

2. Has the statistical analysis been performed appropriately and rigorously?

Reviewer #1: I Don't Know

Reviewer #2: N/A

3. Have the authors made all data underlying the findings in their manuscript fully available?

Reviewer #1: Yes

Reviewer #2: No

4. Is the manuscript presented in an intelligible fashion and written in standard English?

Reviewer #1: No

Reviewer #2: No

5. Review Comments to the Author

Reviewer #1: The manuscript by Mohd Kashif et al. entitled “Recombinant expression and preliminary characterization of Peptidyl-pr?lyl cis/trans-is?merase Rrd1 from Saccharomyces cerevisiae” describes the cloning of the rrd1 gene in pET-21d (+) its overexpression and purification in E. coli.

The authors conclude that the production of the recombinant protein can be used to obtain its biophysical characterization and to identify novel-associatin partners of Rrd1 protein.

I believe, as the authors say, that the production of the recombinant protein can be used to obtain its biophysical characterization and to identify novel-associatin partners of Rrd1 protein, but in my opinion the work should be improved.

I admit this manuscript with major revision.

In particular:

- I would like to suggest that the authors should indicate the yield of purification.

- I would suggest that the authors try to eliminate many repetitive parts in Discussion and Conclusion paragraphs.

- I suggest a revision of the English language.

Other points:

- Is the ampicillin concentration (50mg/mL) correct? May be 50 micrograms/milliliter.

- The resolution of the graphics is very bad. (Figure 1; Figure 3, C and D; Figure S1)

- In the “Overexpression and purification of the recombinant Rrd1 protein” paragraph, the parenthesis is missing (line 3).

- In the caption of Figure 2, opri,ization should be corrected

- The caption of Figure 3, should be corrected.

- The column flow is once declared to be 0.3 and once more 0.4 mL/min.

- The caption of Figure 4 and Figure 5 are not in italic.

- The font of the Figure 5 caption’s is different.

- The numerical sequence of the figures is wrong.

- To check punctuation.

Reviewer #2: The manuscript by Mohd Kashif et al. entitled “Recombinant expression and preliminary characterization of Peptidyl-pr?lyl cis/transis?merase Rrd1 from Saccharomyces cerevisiae” report cloning, expression and purification of the Rrd1 from Saccharomyces cerevisiae, and also an initial characterization. The authors declare that the protein is correctly cloned, expressed in the E. Coli strain, and purified. The authors, by the CD and the Spectro fluorescence, declare also that the protein is correctly folded.

The aim of this work is interesting (and this reviewer appreciates it), the paper is not well done in almost all parts, and the manuscript in the present form demands a deep revision before it can be published, so this reviewer suggests a major revision of the paper.

The paper should be modified in some parts.

Major issues:

1) The abstract needs to be revised according to the following suggestions.

2) The introduction section is too long and dispersive, nothing is reported on the aim of the paper. The results and the methods used should be removed from this section.

3) The section Materials and Methods describe also the results obtained. In this way is difficult to read and understand the paper, please split this section into two sections (e.g. M&M and Results).

4) About the purification is not reported the yield per gram of humid pellet, is not reported a spectroscopic analysis for the purity evaluation of the protein and for the quantization of the protein.

5) In the paragraph “Optimization of recombinant Rrd1 expression and solubility” the most important info is not reported “data not show”, why?

6) Some parts of the computational analysis should be reported before the expression and purification and the rest before the characterization.

7) In the paragraph “Primary sequence analysis and homology modeling of Rrd1” is reported (fig 3a) but it does not match with the text.

8) The paragraph “Structure analysis of Rrd1 protein” should be completely revised or removed. The CD spectra analysis is too simplistic, no spectra post-analysis was done. An investigation in the near-UV should be performed. By the fluorescence analysis performed (just steady-state spectra) is not possible to define the tertiary structure or the right folding of the protein, a deeper fluorescence analysis should be performed to declare the correct folding.

9) As a consequence of the point before also the Discussion and the Conclusions sections need to be revised.

Minor issue:

1) Please revise the text for some misspellings

2) Figure 3 panels c and d are in low resolution, please improve.

3) Figure S1 is in low resolution, please improve it.

---

## [Editor Report · Decision Letter 1]

22 Feb 2023

Recombinant expression and preliminary characterization of Peptidyl-prоlyl cis/trans-isоmerase Rrd1 from Saccharomyces cerevisiae.

PONE-D-22-27039R1

Dear Dr. Mohd Kashif,

We’re pleased to inform you that your manuscript has been judged scientifically suitable for publication and will be formally accepted for publication once it meets all outstanding technical requirements.

Kind regards,

Sabato D'Auria

Academic Editor

PLOS ONE
---

## [Editor Report · Acceptance letter]

26 May 2023

PONE-D-22-27039R1 

Recombinant expression and preliminary characterization of Peptidyl-prоlyl cis/trans-isоmerase Rrd1 from *Saccharomyces cerevisiae*. 

Dear Dr. Kashif:

I'm pleased to inform you that your manuscript has been deemed suitable for publication in PLOS ONE. Congratulations! Your manuscript is now with our production department. 

Kind regards, 

on behalf of

Dr. Sabato D'Auria 

Academic Editor

PLOS ONE